# Application of Real-Time Cell Electronic Analysis System in Modern Pharmaceutical Evaluation and Analysis

**DOI:** 10.3390/molecules23123280

**Published:** 2018-12-11

**Authors:** Guojun Yan, Qian Du, Xuchao Wei, Jackelyn Miozzi, Chen Kang, Jinnv Wang, Xinxin Han, Jinhuo Pan, Hui Xie, Jun Chen, Weihua Zhang

**Affiliations:** 1School of Pharmacy, Nanjing University of Chinese Medicine, Nanjing 210023, China; yanguojun@njucm.edu.cn (G.Y.); XuchaoWei@163.com (X.W.); JinnvWang@163.com (J.W.); hxx0307@163.com (X.H.); Panjinhuo@163.com (J.P.); njxh66@163.com (H.X.); chenjun75@163.com (J.C.); 2Jiangsu Key Laboratory of New Drug Research and Clinical Pharmacy, Xuzhou Medical University, Xuzhou 221004, China; duqian81@163.com; 3Department of Chemical and Biomolecular Engineering, The Ohio State University, Columbus, OH 43210, USA; miozzi.5@buckeyemail.osu.edu; 4Department of Internal Medicine, Carver College of Medicine, University of Iowa, Iowa City, IA 52242, USA; chen-kang@uiowa.edu; 5Beijing Body Revival Medical Technology Co., Ltd., Beijing 100088, China

**Keywords:** RTCA, application progress, traditional Chinese medicine, pharmaceutical evaluation

## Abstract

**Objective:** We summarized the progress of the xCELLigence real-time cell analysis (RTCA) technology application in recent years for the sake of enriching and developing the application of RTCA in the field of Chinese medicine. **Background:** The RTCA system is an established electronic cellular biosensor. This system uses micro-electronic biosensor technology that is confirmed for real-time, label-free, dynamic and non-offensive monitoring of cell viability, migration, growth, spreading, and proliferation. **Methods:** We summarized the relevant experiments and literature of RTCA technology from the principles, characteristics, applications, especially from the latest application progress. **Results and conclusion: **RTCA is attracting more and more attention. Now it plays an important role in drug screening, toxicology, Chinese herbal medicine and so on. It has wide application prospects in the area of modern pharmaceutical evaluation and analysis.

The Real-Time Cell-based Assay (RTCA) technology is a real-time cellular biosensor. This technology allows for uninterrupted, label free, and real time analysis of cells over the course of an experiment. The literature reported that electronic cell plate impedance sensors were used in real-time detection of cellular processes for the first time in 1984 [1]. A series of related studies were then successively produced.

With the development of electronic technology in the field of biological research, RTCA can be more and more widely applied. It is now widely used by researchers around the world in many different fields of research. The RTCA is capable of drug activity screening, drug toxicology research, pathology analysis, etc.

## 1. Principles and Characteristics

### 1.1. Principles

The function of the machine is based on the electron resistance reading of the gold-plated sensor electrode, which is located on the bottom of the plate (E-16 plate) [2] or the lower surface (CIM plate) of the migration plate. The electrical impedance measured by the micro-electronic chip reflects a series of physiological states such as cell growth, stretch, morphological changes, death and adherence. The electrode impedance is mainly determined by the ionic environment of both the electrode interface and the overall solution. As the cells are attached or separated from the surface electrode, the electronic readings change, resulting in impedance changes calculated by complex mathematical algorithms and plotted as cell index (CI) values [3]. In the case that there are more cells on the electrode, the CI value is higher. Moreover, the change in impedance is also related to the breadth (quantity and mass) of cell adhesion to the electrode. Increased cell growth and electrode contact surface will lead to greater changes in impedance. In other words, this impedance reading could be affected by the quality of cell interactions and adherent properties between each cell and the electrodes. Thus, cell biology states including cell viability, cell count, cell morphology, and cell adhesion levels will affect impedance measurements (Figure 1).

The formula for calculating Cl is as follows:$$CI=\underset{i=1\to N}{\max}(\frac{R_{cell}\left(f_{i}\right)}{R_{b}\left(f_{i}\right)}-1)$$
*R*_*b*_(*f*) and *R*_*cell*_(*f*) represent the unimpeded electrical impedance and the electrical impedance of the cell adhesion respectively. N represents the number of frequency points where the electrical impedance is detected. The CI measurement of RTCA can be used for a comprehensive real-time monitoring of cell community changes and has a high degree of consistency and repeatability. The test results are proved to be consistent with other traditional analysis results [4].

### 1.2. Characteristics

For a long time, most of the analysis systems were based on cell screening using markedness tracking to test the function of various cells, such as radioactive isotope, optical absorption, fluorescence, and luminescence. These methods usually need to mark the ligand, enzyme substrates or tracer molecules [5,6,7,8,9]. Applications of the traditional method has great limitations. Marking requires a lot of money and time due to cell cleaning, which can complicate the analysis and physical conditions causing interference. However, the electronic impedance sensor has overcome the above shortcomings and has the advantages of avoiding marks, violation to the cell, and overcoming the interference of the compounds in detection [10,11]. Meanwhile, in the traditional electronic cell impedance sensors (ECIS) system, a great reference electrode and one or more small detection electrodes will be implanted to the bottom of the cell plate for testing. By measuring the electrical impedance between the two kinds of electrodes in real-time we can detect the cell adhesion, extension, and movement. Although the ECIS system can realize real-time, non-invasive, automatic detection of cells, it can only detect a small amount of cells in a specific pore, because pore distribution is not uniform in the cells, making the experimental repeatability poor.

Currently, there are two kinds of the real-time cell electronic sensing system (RT-CES) cell plate specifications, those with 16 orifices and 96 orifices. There is expected to be a design with a smaller 384-well and 1536-well plate. The real-time cell electronic analysis method with the RT-CES system has many advantages. Under the conditions closest to physiological state testing, results are obtained and automatic and continuous detection can perform the whole process of dynamic information gathering giving high accuracy, repeatability, provide a greater dynamic detection range and high information content. The pathway for researchers bring their own, easy to operate software and experimental method is simple, experimental parameters are relatively small, convenient and it allows fast data processing. Someone compared RTCA with two traditional methods, the fluorescent cell activity analysis method and calculation of artificial cell actual quantity of each hole, to detect the cell growth rate [12]. In addition, some scholars also investigated the optimum cell culture conditions and analysis repeatability with RTCA [13]. The experimental results show that the RTCA can be used to ascertain the best cultivating conditions and cell density for the adhesion of cells such as liver cells, myocardial cells, fibroblasts, etc. On the other hand, the decline of cell indices occurs not only because of cytotoxic effect, but also of the existence of other confounding factors. As a result, RTCA can track the cellular growth over the entire period of the experiment. Compared with the endpoint-based methods, using RTCA it is easier to notice the inhibition of cell growth and is especially effective in cell inhibitor research [14,15]. However, the correlation analysis between cell impedance detection and the classic toxic endpoint analysis was confirmed only in limited compounds, so there are some limitations for RTCA. In general, in the study of pharmaceutics, RTCA is still a powerful and reliable tool, because it has the characteristics of being high throughput, rapid, and effective.

## 2. Drug Screening and Development

The traditional drug-screening test uses a marker to quantify biological responses associated with disease states [16,17,18,19]. RTCA can overcome the limitations of this approach; however, to achieve the real-time monitoring of cell state, a powerful tool for drug screening is required. The xCELLigence RTCA Cardio System can dynamically monitor the contractility of myocardial cells, can be sensitive, and can quantitatively detect the effects of myocardial function ion and non-ion channel modulators. Thus, xCELLigence RTCA can be used to evaluate the preclinical cardiac safety of certain drugs and can provide relatively high flux in a convenient way for screening the potential cardiac toxicity of drugs [20]. S. Kustermann [21] and his team used xCELLigence RTCA to measure the cytostasis from cytotoxic drugs by recording time-kinetics of compound-effects on cells. It shows that impedance-based real-time cell analysis could be used to further characterize toxicities observed in vitro as a convenient screening tool. Smout et al. [22] also used the xCELLigence system with a fully automatic and high flux method to monitor in real-time the peristalsis of a parasite. They also determined the difference in real-time of IC50 values to quantitatively evaluate the anthelmintic effect, and simply and objectively evaluated the effect of insect repellent and deworming in drug screening. Abassi et al. [23] used the RT-CES system to monitor the entire process of sensitization and activation of IgE mediated mast cells, for high-throughput screening. IgE mediated inhibitors provide a way to manipulate mast cells. Boitano et al. [24] studied the agonist of PAR2 using xCELLigence RTCA, finding that the new compounds, 1 and 2, are more stable than the agonist that is modified by 2–furan. These new compounds have become the basis of the design of the structure-effect relationship and have laid the foundation for potential the discovery of a powerful agonist and antagonist of PAR2 in the future. Kamran Harati et al. [25] used RTCA to respectively analyze the three different drugs for eight different human soft tissue sarcoma (STS) cell lines. Results showed that the natural compound epigallocatechin-3-gallate (EGCG), reduces all cell proliferation and survival abilities, except for in the 1273 synovial sarcoma cell line, silibinin to synovial sarcoma, liposarcoma and fibrosarcoma cell lines had anti-proliferation effects but noscapine gave no inhibition of proliferation. Therefore, RTCA is a very good method for the screening of STS cell inhibitors. Nguemo et al. [26] evaluated the cause of arrhythmia characteristics of drugs using an in vitro model with xCELLigence RTCA. In the process of research, they combined induced pluripotent stem cells (IPSs), derived from myocardial cells (CMs), with xCELLigence RTCA technology to perform an evaluation of known drugs (isoproterenol, kappa, terfenadine, sotalol and doxorubicin) effects on the activity of the heart. Among them, they used a RTCA Cardio system to analyze the cell reaction after the single dose or repeated administration. Results showed that the induced effect of the related compounds on IPS-CMs fluctuation parameters changed a lot. In addition, the system detected that the existence of doxorubicin led to a serious abnormal phenomena, such as the reduction of the wave frequency and amplitude of IPS-CMs single-layer cells and a change in the pulse signal. The study showed that the xCELLigence RTCA Cardio system, combined with IPS cells to study the myocardial cells, that prolong the incubation time and screen for the cardiac side effects of potential drugs, provides a very attractive high-throughput tool.

## 3. Drug Toxicology Analysis

RTCA has a function of real-time dynamic monitoring of cell proliferation and toxicity changes, so it also can be used in the study of drug toxicity to cells. The study of exogenous factors (chemical, physical, biological and other factors) on biological organ damage and its mechanisms of action is an important aspect of toxicology research. The dynamic cell response map can provide a comprehensive analysis of the toxicity of exogenous substances and its mechanisms of action for exogenous substances providing a comprehensive analysis of the tool. The RTCA technology for early toxicology was studied in the experiment by Xiao et al. in 2002. Xiao et al. [27] studied cytotoxic screening, and they used RTCA to analyze fibroblast V79 cells at the electrodes. In the study, cytotoxicity of cadmium chloride, sodium arsenate and benzalkonium chloride was determined by RTCA analysis, and the standard neutral red method, respectively. From the analysis of the results, as a function of the inhibitor concentration, the percentage inhibition could be seen as that of half the control concentration, the required concentration to achieve 50% inhibition derived from the response function, which agreed well with the results of using the standard neutral red assay. Therefore, it could be derived from the method that chemical cytotoxicity was easily screened by observing the response function of attached cells in the presence of the inhibitor. Measurement and models of diverse information sets are difficult at the analytical level using conventional end point cytotoxicity assays, since current conventional cytotoxicity assays can only detect very specific cellular changes, such as viability, occurring after certain durations of toxicant exposure, which provides no dynamic information with respect to living cells in response to toxicants. Thus, to accurately assess toxicant-induced cellular damage and to understand the mechanism of action for toxic compounds, it is pertinent to examine multiple parameters in the same assay under dynamic conditions. Xing et al. [20] treated cells grown in a micro-electronic sensor with different poisons, such as arsenic trioxide, mercury and sodium dichromate in their cytotoxicity evaluation, and then continuously measured the cellular response to the poisons using the RT-CES system. Then, the IC50 values measured with the RT-CES system were compared with the IC50 values measured by MTT colorimetric assay, lactate dehydrogenase (LDH) release assay and neutral red uptake (NRU) analysis. The results of the RT-CES system was similar to that of NRU at specific time points. In addition, RT-CES dynamically detected the effects of arsenic trioxide, mercuric chloride and sodium dichromate on cells such as NIH3T3, and compared these with the standard labeling endpoint in the toxicity evaluation test. The results showed that the RT-CES system allows for real-time, continuous monitoring and quantitative recording of the whole assay process and provides new insight into cell-toxicant interactions. The toxicology research based on RTCA technology has been widely used in the assessment of environmental toxins, the detection of microbial pathogens and the detection of toxicants in the cosmetics industry. Qi Zhao et al. [28] established an early screening method for cardiotoxicity in vitro, combining human embryonic stem cell derived cardiomyocytes (hESC-CM) with real-time cell analysis cardio (RTCA Cardio). In other aspects, Zhang et al. [29] used RTCA to analyze the in vitro cytotoxicity of different concentrations of APP (antibacterial of persistent photocuring) antimicrobial agent in order to evaluate the biosafety of the antibiotic, preliminarily. The antimicrobial agents were prepared in 10%, 5%, 2.5%, 1.25%, 0.625%, 0.31% and 0.15% dilutions. The results demonstrated that 2.5%, 5%, and 10% dilutions of the antibacterial agent APP have a strong inhibitory effect on the growth of human oral cancer KB cells. Medical devices may be in touch with human tissues and cells directly or indirectly and thus need biocompatibility. A cell toxicity test is an important indicator for toxicity evaluation of medical devices. Wang and Xu investigated the linear range, precision and reproducibility of RTCA by using L929 cells and compared the method with the MTT colorimetric method. As a result, the RTCA method was found to be suitable for the evaluation of the in vitro cytotoxicity of medical devices [30]. Caixia Yang [31] used human umbilical type II epithelial A549 cells, human normal lung epithelial BEAS-2B cells, human non-small cell lung cancer H1975 and HCC827 cells in their research model, they employed cell morphology observations, RTCA real-time unlabeled cell analysis, and WST-1 technology to study the biological effects of nano-zinc oxide on normal and tumor cell lines of various respiratory systems. At the same time, the RTCA real-time cardiomyocyte analysis technique was used to explore the toxic effect of nano-zinc oxide on primary neonatal rat cardiomyocytes and cardiomyocyte beating. The study found that in the concentration range of 12.5 μg/mL to 100 μg/mL, nano-zinc oxide toxic effects on normal and tumor respiratory system cell lines are dose-dependent. At the same time, the results of the study confirmed that in the concentration range of 12.5 μg/mL to 100 μg/mL, nano-zinc oxide had an effect on primary neonatal rat cardiomyocytes vitality, beating amplitude and frequency.

## 4. Pathology Analysis and Other Studies

RTCA is the most basic application in cell growth, division, proliferation and death mechanism research. Some scholars use the RTCA system for dynamic monitoring of the cytopathic effects caused by influenza A (H1N1) virus and directly compare the results with a traditional optical microscope, where the CI value obtained by the system can reflect the growth state and cytopathic effect. Dowling et al. [32] used the RTCA to real-time monitor HCT116 cell adhesion, proliferation and migration under the influence of fibroblast media. The results showed that in the presence of the medium, of the HCT cells invading the human basement membrane layer, the existence of fibroblast medium could increase their mobility and aggressiveness. Fang et al. [33] and his team used the RTCA technique of to detect the cytopathic effects of Vero cells infected with West Nile virus (WNV) and St Louis encephalitis virus (SLEV). Analysis of the results showed that the cells with WNV infection showed pathological changes faster than the cells with SLEV infection. The mathematical model based on the dynamic curve of the lesion effect of impedance shows that the multiplication rate of WNV was three times faster than that of SLEV. In addition, the RTCA system was used to quantitate the cell protection levels produced by neutralizing antibodies against WNV and SLEV. In this study, RTCA provided a high-throughput and quantitative method for real-time detection of virus proliferation and neutralizing antibody inhibiting effects. Ryder et al. [34] studied the application of the RTCA system in the evaluation of *Clostridium difficile* toxin in human feces by dynamic reaction for measuring the effect of tcdB toxin on human dermal fibroblast cell line HS27. They took 300 patients with suspected *C. difficile* infection using feces as experimental samples, each sample was analyzed by real-time PCR in turn, double glutamate dehydrogenase-toxin ELISA and RTCA analysis. The analyses showed that RTCA analysis had a specificity of 99.6% and sensitivity of 87.5%, which was higher than the ELISA results, but lower than the real-time PCR results. The results also showed that RTCA analysis provided an effective method to evaluate *C. difficile* infection. Ramis et al. [35] analyzed the differences between serum and plasma cell lysis with anticoagulants using xCELLigence RTCA. Results showed that heparin treated plasma produced complement inhibition and underestimate the cytolytic reaction, while EDTA treated plasma resulted in the death of most cultures, suggesting that EDTA is not suitable for the analysis of plasma samples, it seemed that the most accurate sample is serum. At the same time, they also detected the heparin sodium concentration of the antibody and complement mediated cytolysis effect using the RTCA method. The results showed that adding heparin did not reduce antibody and complement mediated cytolysis [36].

Some scholars respectively use RTCA impedance and transepithelial electrical resistance as the output of barrier characteristics to analysis whether the small GTP Rap1 protein is able to adjust the connection of retinal pigment to the epithelial barrier [37]. In the process of studying the redox-specific APE1 inhibitor, APX3330, limited retinal angiogenesis in vitro, Jiang et al. [38] chose to use APPX3330 processed retinal vascular endothelial cells as the research focus, carrying out the proliferation test, transmembrane migration experiment, the basement membrane tube forming experiment and RTCA experiment using the xCELLigence system. The RTCA experimental results showed that wild type RVECs (retinal vascular endothelial cells) and the RVECs with removed VLDL-r (very-low density lipoprotein receptor), after dealing with the APPX3330, the cell index fell. Among these, the latter were more sensitive to APPX3330 concentration changes. This selective effect will help in future therapeutic applications. In drug research, G protein coupled receptors (GPCRs) play an important role in the treatment of cancer, inflammation, and cardiovascular diseases, as they are considered potential drug targets. Yu et al. [39] used electronic cell sensors to monitor the morphological changes of cells in real time and used RTCA for the study of GPCRs. GPCRs activation caused cells to change form. These changes could be accurately detected by real-time sensitive electronic analyzer, which could be used for the physiological evaluation function of GPCR.

RTCA also provides a great method to follow the effects of gene silencing on cell survival. Gebert et al. [40] used the xCELLigence RTCA system for real-time monitoring of cell viability. Immortalized human bronchial epithelial cells (16HBE14o-) were used as an unfolded protein response (UPR) cell fate model to determine UPR-related apoptosis induced by pharmacological endoplasmic reticulum (ER) stress. From the results, they proposed that two hours represented the initial adaptive response to ER stress, six hours corresponded to cell fate decisions, and nine hours represented the start of the UPR-related apoptosis in the 16HBE14o- cells. The RTCA application provided an initial discovery of the mRNA levels for both pro-survival and pro-apoptotic UPR reporters.

## 5. Chinese Herbal Medicine and Natural Product Research

Due to the diversity and variability of constituent components, the complexity of targets and mechanisms, and the unpredictability of the interactions between components being integral to traditional Chinese medicine, it is difficult to evaluate its quality [41,42,43]. With concern to RTCA applications, some want to apply it to Chinese herbal medicine research. Fu et al. [44] studied a novel phenotypic assay based on cell-impedance, using electron impedance to detect cell responses to Chinese herbal medicine in a timely fashion. By comparing the time/dose-dependent cell response profiles (TCRPs) produced by *Cordyceps sinensis* with similar TCRPs produced by chemical drugs, it can be concluded that TCRPs of Chinese herbal medicine have a potential application in predicting cellular mechanisms, the identification of Chinese herbal medicines, and the identification of biological activity.

In addition, hoping to provide a new method for the study of traditional Chinese medicine quality standards, Pan et al. have used RTCA on traditional Chinese medicine, like *Hirudo nipponica Whitman* [45], *Panax notoginseng* [46] (Figure 2), Compound Danshen Dripping pills [47], etc. For the past several years, RTCA has been used to explore the biological activity of more Chinese medicine.

Sheng and Shi [48] compared the biological activity of different parts of *Tripterygium wilfordii* based on real-time cell analysis combined with HPLC. The results showed that the activity of *T wilfordii* was similar at the same site. The root bark and leaves of *T. wilfordii* had similar biological activity, and the roots and stems had similar activity. The biological activity of basophilic leukemia 2H3 (RBL-2H3) cells in root bark and buds was significantly higher than that in the roots and stems, especially in the root bark, where biological activity was the strongest. The fingerprint technique used, showed that the components of different parts were similar, but the contents were largely different. Gao and Bi [49] employed an MTT assay, real-time cell analysis and high-content cell imaging systems for detecting the proliferation, migration and expression of F-actin in A549 cells under different conditions. They found that a water extract of ginseng could effectively inhibit the proliferation and migration ability of A549 cells in co-culture system, which may affect the biological behavior of tumor cells by regulating the immune activity of tumor associated macrophages (TAMs).

Yan et al. [50] built a novel method for the evaluation of release kinetics for Chinese medicine compounds: They applied the xCELLigence RTCA system to determine the release characteristics of sustained-release pellets containing *Sedum sarmentosum* compounds (Figure 3). They concluded that the new method of cell-index release kinetics might provide a quantitative description for the release of multi-active agents from traditional Chinese medicines. The application of the xCELLigence RTCA system to evaluating the release kinetics of Chinese medicine compounds is feasible. In the latest study about real-time cell analysis for traditional Chinese medicine, Lv et al. also applied RTCA when they studied effective components from *Gentianella acuta* [51]. The RTCA is not only used for studying traditional Chinese medicine, but more and more natural product research makes use of this technology. Bartoszewski et al. [52] and his team used RTCA to evaluate tea extracts (*Cyclopia* sp.) on cervical cancer (HeLa) cells, they found that the higher hesperidin content in non-fermented “green” extracts correlated with higher cytotoxicity compared to fermented extracts. They also found that mangiferin had a modulatory effect on the apoptotic effects of hesperidin with real-time cellular cytotoxicity analysis. Therefore, RTCA technology can provide a promising method for traditional Chinese medicine and natural product research.

## 6. Perspective

The RTCA system is based on the measurement of electrical micro-impedance through gold-plated sensor electrodes, so the detection cost may be higher than other devices. In addition, CI values are detected through adherent cell cultures, which are two-dimensional (2-D) cellular cultures. It may lack three dimensional scaffolds to support cell growth and proper tissue functions, and cannot mimic in vivo micro-environments [53]. On the other hand, if cells are grown in suspension, RTCA cannot generate any significant change in the micro-impedance signal. But Obr et al. [54] reported that they could measure the hematopoietic cells growing in suspension with embedded micro-electrodes coated with a cell-binding protein fibronectin fragment (FNF). The micro-impedance signal specifically reflected cell binding to the coated surface. The results indicated that the role of Src kinases in the regulation of hematopoetic cell adhesion signaling is similar to that of c-Src in adherent cells.

In spite of its limitations, RTCA is attracting more and more attention. Now it plays an important role in drug screening and development, toxicology analysis, pathology analysis, Chinese herbal medicine research and so on. It has a wide application prospect in the area of modern pharmaceutical evaluation and analysis, especially in traditional Chinese medicine.

Through the screening of one or a variety of traditional Chinese medicines with specific cell lines, a dynamic model could be built to establish a cell index resource. The model has the character of uniqueness, repeatability, and reversibility for certain compound aggregates. The RTCA model has the potential value for the identification and quantitative analysis of unknown compound mixtures. If a correlation between cell indices and effectiveness of traditional Chinese medicines could be built, there could be more significance on quality evaluation of traditional Chinese medicines. It is believed that combined with other advanced chemical and biological technologies, RTCA technology can provide a promising method for the research of traditional Chinese medicine and compounds.

## Figures and Tables

**Figure 1 molecules-23-03280-f001:**
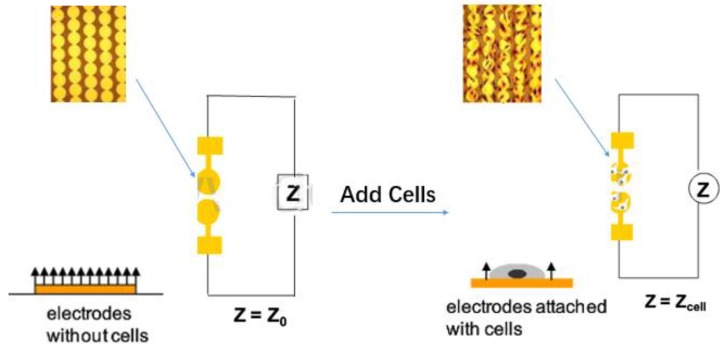
Schematic diagram for the mechanism of Real-Time Cell-based Assay.

**Figure 2 molecules-23-03280-f002:**
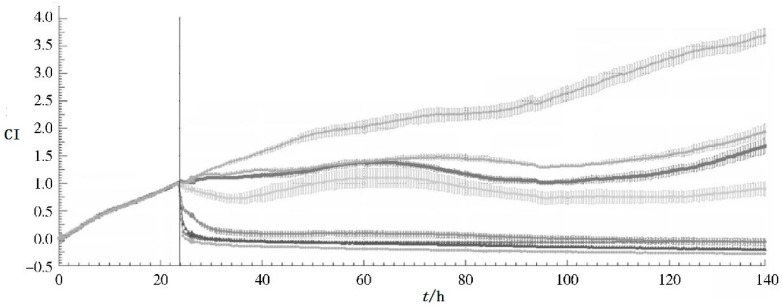
CI-t curve of MCF-7 cell line based on different concentration of *Panax notoginseng* extractive (curve from up to bottom show the concentration as con, 0.17, 0.28, 0.47, 0.78, 1.3, 2.16, 3.6 mg/mL).

**Figure 3 molecules-23-03280-f003:**
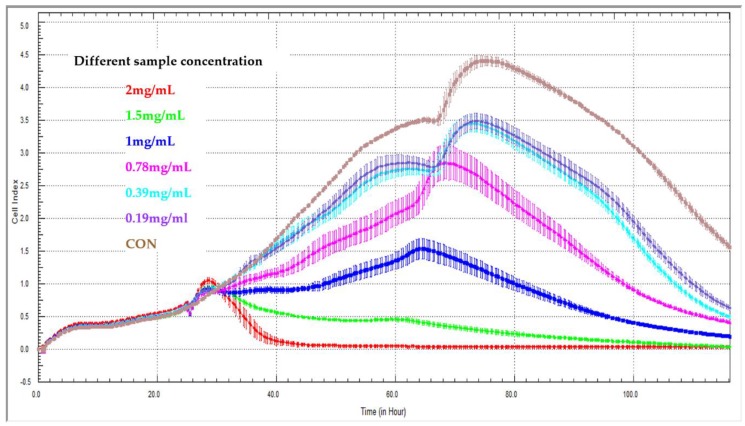
Effect of different sample concentrations on real-time cell analyzer curves generated with LX2 cells. Effect of 2 mg/mL (red line), 1.5 mg/mL (green line), 1.0 mg/mL (blue line), 0.78 mg/mL (pink line), 0.39 mg/mL (turquoise line), 0.19 mg/mL (purple line), CON (brown line, control group without treatment).

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
