# Peer review of "Application of Real-Time Cell Electronic Analysis System in Modern Pharmaceutical Evaluation and Analysis"

_molecules, 2018, doi:10.3390/molecules23123280_

Round 1

Reviewer 1 Report

Dear Authors,

the manuscript describes the application of real-time cell electronic analysis system and has some merit. However, the manuscript is poorly written, there are many typographical errors, typos, grammatical errors, lack of references (for example - line 121, lack of authors names - line 360). Fig. 2 and 3 are not mentioned in the text.

The references in the text is written inconsistently. The text is also incomprehensible and incoherent in many places.

Any abrreviations should be described (for example: TCRPs).

In the section: "Drug screening and development" is the mix of different systems - it should be better organized.

Furthermore, please explain: 1. the usefulness of rtca in vivo studies? (line 111), 2. the sentence in line 182-183 (the RTCA method is suitable for.....), 3. lines161-168 - the text is unclear.

In summary, the manuscript should be extensively improved, better organized and precisely checked.

Author Response

Point 1The manuscript describes the application of real-time cell electronic analysis system and has some merit. However, the manuscript is poorly written, there are many typographical errors, typos, grammatical errors, lack of references (for example - line 121, lack of authors names - line 360). Fig. 2 and 3 are not mentioned in the text.

Response 1: We are so sorry to lack of whole check for the manuscript; we have tried to amend all these mistakes(such as line 121, Fig.2 and 3,etc.), In regard of reference 24- line 360, we found the journal was suspension of publication and the article was truly lack of authors, we deleted this reference. Thanks a lot.

Point 2The references in the text is written inconsistently. The text is also incomprehensible and incoherent in many places.

Response 2: We apologized that we have you in trouble; we neaten the references with EndNote and try our best to reorganization the text. Thank you so much.

Point 3Any abrreviations should be described (for example: TCRPs).

Response 3: We checked the manuscript and made the modifications. Thanks a lot.

Point 4: In the section: "Drug screening and development" is the mix of different systems - it should be better organized.

Response 4: In the section of "Drug screening and development", different RTCA systems were summarized to illustrate the application of high-throughput screening of RTCA. We tried to fix the typos and mistakes and make the expression clearly.

Point 5: Furthermore, please explain: 1. the usefulness of rtca in vivo studies? (line 111), 2. the sentence in line 182-183 (the RTCA method is suitable for.....), 3. Lines161-168 - the text is unclear.

Response 5:

The application of RTCA is mainly used for detection of cells, so in vivo studies is not included, we deleted it.

The RTCA method is suitable for the evaluation of in vitro cytotoxicity of medical devices. Medical devices may be in touch with human tissues and cells directly or indirectly and it needs biocompatibility. Cell toxicity test is an important indicator for toxicity evaluation of medical devices. We added these explainations in the manuscript.

We have tried to amend the lines161-168 of original manuscript and make it sense clearly.

Thanks so much for all these suggestions.

Reviewer 2 Report

This is an interesting  manuscript. However I have couple of comments.

Figures quality should be improved.

Not only chinese plant extracts were studied using RTCA: please see for example :

https://doi.org/10.1371/journal.pone.0092128

3. RTCA provides great method to follow effects of gene silencing on cell survival - this should be disccused :

please see. https://doi.org/10.1038/s41598-018-34861-2

4. RTCA gives ability to assay cells in suspension - this should also be discussed.

5. Finaly although this method is really nice, it also has numerous limitations that should be clearly stated in conclusion section

Author Response

Point 1 Figures quality should be improved.

Response 1: We have tried our best to make the figures more clearly, because the figures are from original references, we just can take limited adjustment for the pictures. Hope they can express the meaning schematically. Thanks so much for good suggestions.

Point 2 Not only chinese plant extracts were studied using RTCA: please see for example :

https://doi.org/10.1371/journal.pone.0092128

Response 2: Thanks for good suggestion and we added this reference in the manuscript.

Point 3. RTCA provides great method to follow effects of gene silencing on cell survival - this should be disccused :please see. https://doi.org/10.1038/s41598-018-34861-2

Response 3: Thanks for good suggestion, we discussed this application of RTCA and added the reference in the manuscript.

Point 4. RTCA gives ability to assay cells in suspension - this should also be discussed.

Response 4: Thanks for enlightenment and we added references about this theme in the6. Perspective” of manuscript.

Point 5. Finaly although this method is really nice, it also has numerous limitations that should be clearly stated in conclusion section

Response 5: We added the discussion of the limitations at the beginning of “6. Perspective”, Thanks a lot for good suggestions.

Round 2

Reviewer 1 Report

Dear Authors,

the manuscript has many errors, for example:

-line 43 - "on the cytoplasmic plate" should be delete,

-line 71 - should be "however",

--line 87 - should be "compared",

-line 91 and 92 - the sentence is confusing,

-line 96 - what is "I" in the sentence?,

-line 100 - please check commas,

-line 110 - should be "Kustermann",

-"in vitro" is written in italics or without italics. It should be unified, similarly in the case of names of plant species or microorganisms,

-some of the abbreviations are still not explained: for example line 178 - APP or line 284 - TAMs,

-in the description of Fig.2  - what is "con"?

-names of authors are written in a different way.

Please, check again your text carefully.

Author Response

Point 1-line 43 - "on the cytoplasmic plate" should be delete,

Point 2-line 71 - should be "however",

Point 3-line 87 - should be "compared",

Point 4-line 91 and 92 - the sentence is confusing,

Point5-line 96 - what is "I" in the sentence?,

Point 6-line 100 - please check commas,

Point 7-line 110 - should be "Kustermann",

Response 1-7: We are so sorry for the above mistakes; we corrected them and checked the whole manuscript seriously. We amended the expression from line90-102; hope to make sense clearly. Thanks very much, for your careful checking.

Point 8-"in vitro" is written in italics or without italics. It should be unified, similarly in the case of names of plant species or microorganisms,

Response 8: We unified the italics of “in vitro” and “in vivo”, and names of plant species or microorganisms, Thanks a lot for your good advice of modification.

Point 9-some of the abbreviations are still not explained: for example line 178 - APP or line 284 - TAMs,

Response 9: We have try our best to expand most of the abbreviations in manuscript. Thanks very much for suggestions.

Point 10-in the description of Fig.2  - what is "con"?

Response 10: We guess you mean “con” in Fig 3, the “con” means control cell group without any treatment. We explain it in figure representation, Thanks very much for good question.

-names of authors are written in a different way.

Response 5: We checked the names of authors and made some modification as commas, the way of written, and so on. Thank you very much for careful checking.

Reviewer 2 Report

The majority of my comments were addressed. However, couple of minor comments

1. please change in line 244 " Magdalena's Gebert group" into "Magdalena Gebert and coworkers."

Author Response

Point 1. please change in line 244 " Magdalena's Gebert group" into "Magdalena Gebert and coworkers."

Response 1: We changed it as your suggestion, Thanks so much for your help.